# Review and Proposal for a Classification System of Soft Robots Inspired by Animal Morphology

**DOI:** 10.3390/biomimetics8020192

**Published:** 2023-05-04

**Authors:** Alexandro López-González, Juan C. Tejada, Janet López-Romero

**Affiliations:** 1Department of Engineering Studies for Innovation, Universidad Iberoamericana, Ciudad de México 01219, Mexico; alexandro.lopez@ibero.mx (A.L.-G.);; 2Computational Intelligence and Automation Research Group (GIICA), Universidad EIA, Envigado 055428, Colombia

**Keywords:** soft robotics, bioinspiration, classification, animal morphology

## Abstract

The aim of this article is to propose a bio-inspired morphological classification for soft robots based on an extended review process. The morphology of living beings that inspire soft robotics was analyzed; we found coincidences between animal kingdom morphological structures and soft robot structures. A classification is proposed and depicted through experiments. Additionally, many soft robot platforms present in the literature are classified using it. This classification allows for order and coherence in the area of soft robotics and provides enough freedom to expand soft robotics research.

## 1. Introduction

Since the inception of the robotics discipline, it has drawn massive inspiration from biology, imitating animal capabilities to provide robotic systems with new forms of actuation and sensing, as well as new modeling and control techniques [1].

The amazing animal kingdom, through evolution, has endowed animals with interesting and special capacities for survival. Their continuous adaptation to a changing environment gives them advantages in locomotion, cognition, control, and information processing. For example, a mouse or an octopus can pass through a small hole by squeezing its deformable body through it; it is capable of maneuvering through confined spaces without suffering pressure or stress concentration damage. Another remarkable example of extreme capabilities involves froghoppers, which are able to jump up to 70 cm vertically, accelerating at 4.000 m/s^2^ and experiencing over 400 g—unit of acceleration, without damage and with low energy consumption.

All living beings, including humans, are mainly composed of soft and flexible materials, such as muscles, skin, and even bones. In contrast, traditional robots have been developed with hard actuators and rigid structures, which are effective mainly in industrial environments and applications. However, in unstructured environments where robots need to interact with humans or animals, they pose risks and limitations due to differences in structural softness and actuator elasticity [2].

Soft robotics is based on mimicking the structures, sensors, and actuators that are present in animals, such as worms, snakes, jellyfish, octopuses, and many others [3]. Soft structures, sensing, and the movement strategies of these animals in complex environments represent desirable advantages [4,5]. As an example, earthworms and inchworms use peristaltic movements to move in a single axis, which may be beneficial for the exploration of human intestines and gastroesophageal cavities [6].

Soft robotics is an emerging field with a wide range of materials, platforms, actuators, sensors, and control strategies. As such, it can be challenging to navigate and organize the diverse advancements made in the field, resulting in fragmented and disorganized landscapes. However, the authors of this work propose a solution to this problem in Section 4. We suggest that a bio-inspired classification could be used to structure the study area of soft robotics, which would help to systematize the various advances in the field and generate a better understanding of the science behind the technology.

The proposed bio-inspired classification scheme has the potential to bring order to the field of soft robotics and facilitate the development of new and innovative soft robotic technologies. By grouping advances into structures and actuation categories based on their morphology, researchers can more easily identify areas of opportunity for further exploration and development. This classification scheme may also allow for a more straightforward comparison of different soft robotic technologies, enabling researchers to assess their relative strengths and weaknesses in a more meaningful way.

This paper is organized as follows: Section 2 expands on soft robot definitions, Section 3 describes biological beings from a structural and actuation viewpoint, and in Section 4, the proposed classification is described and depicted using several experimental platforms. They are also shown other authors’ soft robots classified using the proposed bio-inspired classification. Lastly, Section 5 presents the advantages of the classification, reflections on the topic of soft robotics, and conclusions.

## 2. Soft Robots

Soft robots are constructed using soft and deformable materials, and their softness is typically defined in terms of elasticity or Young’s modulus [7,8,9]. Although this may not be a perfect way to describe softness [8], it highlights the reason for the existence of soft robots. Conventional robots are typically made of metals and plastics with moduli in the order of 10 GPa to 10 TPa, whereas natural organisms are composed of skin and muscles with moduli in the order of $10\;\times \;10^{4}$ Pa to 10 GPa. This large difference in elasticity between materials can lead to safety concerns when in contact, with uneven force distribution and stress concentration potentially causing damage to the softer material, which is typically biological tissue [7].

As different authors have proposed their own definitions of soft robotics, it is important for this work to consider the definition proposed by Wang and Iida [9], “*We define soft-matter robotics as robotics that studies how deformation of soft matter can be exploited or controlled to achieve robotic functions*”.

Lashi and Cianchetti [10] propose a definition of soft robotics that goes beyond a matter-based approach. They describe soft robotics using two approaches, “*control of the actuator stiffness of robot with rigid links” and “softness intrinsically due to the passive characteristics of the robot bodyware*”.

Another definition of robot softness and soft robotics is given by Chen and his colleagues [11]: “*Softness of a robot is thus defined as: the stress and other damage quantities created in a robot’s environment as well as receiver given a particular material deformation in a particular structural configuration of the robot*”, and “*Soft robotics is the subject to study how to make use of the softness of an object or a piece of materials or a system for building a robot by satisfying a required softness to both its environment and its receiver*”.

In 2016, the RoboSoft community defined soft robots as “*soft robots/devices that can actively interact with the environment and can undergo “large” deformations relying on inherent or structural compliance.*” For more definitions of soft structures, soft control, and other types of soft robot compliance, we recommend reading the article by Wang, Nurzaman, and Iida [12].

Soft robots have a great diversity of structures, materials, and actuation [13,14,15]. However, soft robot structural characteristics, actuators, and materials share design coincidences. For example, platforms and actuators created with silicone, such as ecoflex or dragonskin, and actuated by means of pressurized air, have common structural elements, including solid silicone walls for support and inflatable cavities for actuation. Soft robots take inspiration from nature, where there are also coincidences in structural design, materials, and actuators among groups of animals. The next section provides an analysis of animal morphology.

## 3. Morphology Analysis

Multi-cellular animals can be divided into two groups: invertebrates and vertebrates. The spine is the component that characterizes a vertebrate animal. Mammals, reptiles, birds, and fish have spines and, therefore, belong to the vertebrate group. On the other hand, worms, sponges, insects, spiders, starfish, and other similar organisms do not have a spine, and belong to the invertebrate group. Examples of invertebrates and vertebrates can be seen in Figure 1 [16].

### 3.1. Animal Support Structures

Both invertebrates and vertebrates require a skeleton for support, movement, and protection. A skeleton is a solid or fluid system that allows muscles to return to their original length after contraction [17].

For vertebrates, the most common skeletal system is the endoskeleton, while for invertebrates, the exoskeleton and the hydrostatic skeleton prevail. Many animals have more than one type of skeleton. For example, a tortoise has both an endoskeleton and an exoskeleton [18].

The endoskeleton forms deep within the body; it includes fibrous connective tissue, bone, and cartilage. It is actuated using agonist and antagonist skeletal muscles attached to bones by tendons, as depicted in Figure 2a.

The exoskeleton is formed from within the dermis and epidermis. It varies in thickness and is not uniformly hardened over the entire body, which makes many regions thin and flexible, forming joints. Appropriate agonist and antagonist muscles allow a jointed exoskeleton to move, as shown in Figure 2b.

The hydrostatic skeleton includes a fluid-filled cavity enclosed within a membrane, usually encased with a muscular layer, as shown in Figure 2c. At its simplest, the muscular layer is composed of circular and longitudinal bands of muscle. Contractions of the circular muscles lengthen the organism’s body, while contractions of the longitudinal muscles shorten the body, as depicted in Figure 2d. The fluid, mainly water, must be incompressible for the hydrostatic skeleton to function properly [19].

### 3.2. Animal Actuation

Muscles provide animals with their main source of actuation. There are two general classes of muscle named for the characteristic appearance of individual cells: striated and smooth. The muscles that move and support the skeletal framework are made of striated muscle cells, as shown in Figure 2. Smooth muscle is found in the gut, blood vessels, the uterus, and other locations where contractions are usually slow. The cells that make up the muscle of the heart are also striated like skeletal muscles, but they are electrically different and are usually regarded as a distinct class of muscle [20].

Animal inflation is a noteworthy actuation mechanism. For defensive purposes, some animals, such as the pufferfish, can inflate their bodies up to triple their volume by pumping water into their stomach [21]. The pufferfish and other animals exhibit striking structural and functional specializations for inflation, including large and extensible cavities, the absence of bones, highly stretchable skin, and specialized musculature. Another aspect of inflation is seen through the mantle cavity present in all mollusks, which allows locomotion through jet propulsion [22]. Nautiluses, squids, and octopuses move rapidly by expelling water from the mantle cavity through a tube called a siphon using quick muscle contractions. For the mantle cavity to act as a jet propulsion actuator, it needs to be attached to a shell or be thick and replete with muscles.

### 3.3. Support Structure with Actuation

The organ known as the muscular hydrostat serves as both a support structure and an actuator. It mainly consists of muscles with no skeletal support and has a composition similar to the hydrostatic skeleton in the sense that both use water (muscle tissue itself is mainly made of water), which is incompressible at physiological pressures, to function. However, the water cavity surrounded by muscles in the hydrostatic skeleton can provide support to other structures, while the muscle hydrostat cannot [23]. Examples of muscular hydrostats include elephant trunks, mammal tongues, and cephalopod arms, which are mainly composed of muscle tissue, as shown in Figure 3a,b.

## 4. Soft Robot Classification

One of the main justifications for a soft robot classification is the analogy with other well-studied and heavily researched robots. For example, robotic arms are classified by the type, number, and arrangement of rotational and prismatic joints [24], while parallel robots are classified using similar parameters [25]. Wheeled mobile robots are classified based on the number and type of wheels and their degree of mobility [26], and legged robots are classified by the number of legs and degrees of freedom of the legs [27]. These classifications provide general methodologies for modeling and control, such as the Denavit–Hartenberg [24] or the Canudas de Wit [26] kinematic modeling.

The variety of soft robots demonstrates the difficulty of modeling and control [28,29]. Therefore, a classification could pave the way for modeling and control methodologies. Soft robots are usually classified by their parts, actuation, structure, materials, and sensors [2,30]. While separating a soft robot into its parts or subsystems is an intuitive way of studying its behavior, the biological inspiration that motivated the creation of soft robots shows us that mixed materials and interconnected systems generate exceptional performance and coordination of the systems.

### 4.1. Bio-Inspired Classification

Biological inspiration has been a major driving force in soft robot research. Therefore, we propose using animal morphology as a guide to classify soft robots. As detailed in Section 3, three skeletal systems and three animal actuation mechanisms serve as sources of inspiration for soft robots: endoskeleton, exoskeleton, hydrostatic skeleton, skeletal muscles, muscular hydrostats, and inflation. Table 1 presents the proposed classification, which uses two defining characteristics for soft robots: support structure and actuator type.

As the support structure, we consider the endoskeleton, which must be composed of a hard structure surrounded by a soft material; the exoskeleton, which must have a hard structure outside with soft material inside or a hard structure with weaker sections acting as joints; the hydrostatic skeleton, which uses a fluid (or soft fluid-like material) as the support structure; and the muscular hydrostat, a structure composed of soft actuators.

For actuators, we consider skeletal muscle actuators that provide linear agonist–antagonist movements and inflatable actuators that use a work fluid to transmit power or deformations. This type of classification has the capacity to integrate different actuators. For example, cable-driven, shape memory alloy, magnetic-based, spring, and other related actuators could be classified as muscular actuators. On the other hand, pneumatic, hydraulic, chemical, and other related actuators could be classified as inflatable actuators.

### 4.2. Prototypes

To better exemplify the possible combinations for the proposed classification, we developed eight soft robots as shown in Table 2. Each prototype was developed to detail the type of classification and is associated with a table detailing various classified soft robots from the literature.

#### 4.2.1. Endoskeleton Soft Robots

Figure 4 and Figure 5 show soft robots from the classes EndMu and EndIn, where the endoskeleton is covered with soft tissue. The EndMu prototype is represented by the schematic diagram in Figure 4a and the prototype in Figure 4b, which uses a PLA 3D-printed articulated fish as the endoskeleton covered by Ecoflex 35 and is actuated by a nylon-coated wire. The prototype moves its tail by pulling the cable as an agonist movement, and the antagonist movement is provided by the Ecoflex returning to its original form.

The prototype EndIn is represented by the schematic diagram in Figure 5a and the prototype in Figure 5b. It uses a PLA 3D-printed articulated section as an endoskeleton covered by Ecoflex 35 and actuated by pneumatic chambers within the Ecoflex. The prototype moves using pneumatic inflation, some examples are shown in the Figure 6.

Table 3 lists literature examples of EndMu and EndIn classes, such as the robot elephant’s trunk manipulator by [31], the soft humanoid robotic hand by [32], the spherical rolling robots [33], and the peristaltic crawling robot [34].

#### 4.2.2. Exoskeleton Soft Robots

Figure 7 and Figure 8 display soft robots of classes ExoMu and ExoIn, where the exoskeleton protects a soft tissue and weaker sections act as articulations. The ExoMu prototype is represented by the schematic diagram in Figure 7a, and the prototype is shown in Figure 7b. It is built with a 3D-printed TPU (black) core, protected by 3D-printed PLA (blue) sections acting as the exoskeleton, and the inner part is made of soft polyurethane foam (white). The prototype is actuated by pulling a nylon-coated wire as an agonist movement, while the TPU and the polyurethane foam provide the antagonist movement by returning to their original form.

The prototype ExoIn, represented by the schematic diagram in Figure 8a, and the prototype in Figure 8b, are similarly built to the prototype ExoMu, but without the polyurethane foam-filling. A pneumatic fluidic actuator (twisting balloon) is used to expand the articulation.

The exoskeleton soft robot configurations classified in group 2, shown in Table 4, mainly consist of origami robots and compliant robots with soft actuators. Although origami robots obtain their name from the Japanese art of paper folding, their structural similarity to insects is striking: the act of folding paper (or other materials) creates a weaker, softer joint at the crease or pleat, similar to an insect’s exoskeleton. Some of the features observed in these robots include the ability to fold and unfold from external signals [48] as well as the ability to adapt to small spaces and resist external loads [49]. Some examples are shown in the Figure 9.

#### 4.2.3. Hydrostatic Skeleton Soft Robots

Figure 10 and Figure 11 depict soft robot classes HySMu and HySIn. The HySMu class is shown in the schematic diagram in Figure 10a and the prototype in Figure 10b shows a prototype 3D-printed entirely of TPU, actuated by a monofilament nylon fishing line. Being built of just one soft material, with no exoskeleton or endoskeleton, the bottom continuous part acts as a hydrostatic skeleton, where the cable creates the agonist movement, and the antagonist movement is provided by the TPU hydrostatic skeleton returning to its original form.

Secondly, the HysIn class is represented by the schematic diagram in Figure 11a and the prototype in Figure 11b. It uses Ecoflex 35 to create a hydrostatic skeleton and a series of pneumatic chambers on top of it. The inflated chambers generate bending motions, while the continuous bottom part acts as a hydrostatic skeleton providing support to the pneumatic chambers.

The largest classification, shown in Table 5, may be due to the biological influence on soft robots, as well as the basic and well-studied examples such as octopuses, worms, and other invertebrates. In this classification, we can find many robots with pneumatic actuators, as shown in the Figure 12, classified as HySIn, such as those in [76] or [77]. Another interesting and unusual example is the Jellyfish 2D muscle architecture robot [78], which simulates a jellyfish actuated by rat cardiomyocytes.

#### 4.2.4. Muscular Hydrostat Soft Robots

Soft robot classes MuHMu and MuHIn are presented in Figure 13 and Figure 14. The cable-driven muscular hydrostat is represented by the schematic diagram in Figure 13a and the prototype in Figure 13b. It is built using three cables inside a cable housing (such as those used in bicycle V brakes), attached to a PLA triangular end part, and is actuated by pulling the cables in a differential manner. This configuration is unique because no structural elements are used; the actuator functions as the structure, as in mammal tongues or elephant trunks.

The fluidic muscular hydrostat is presented by the schematic diagram in Figure 14a and the prototype in Figure 14b, where two McKibben artificial muscles provide support and actuation to the PLA 3D printed base. Differential pneumatic actuation of the muscles provides movement for the robot.

Examples of muscle hydrostat soft robots from the literature are presented in Table 6, including octopus-inspired arm prototypes [150,151], a peristaltic robot made of cables inside a housing [152], and miniature aquabots [153]. These examples are shown in the Figure 15.

### 4.3. Classification Analysis

The proposed classification scheme offers an overview of the development of soft robots, providing a clear and concise way to define the requirements and characteristics of different types of soft robots. In engineering design processes, properly defining the requirements and characteristics is essential, as without it, the range of possibilities can become overwhelming. By establishing four types of soft robots with distinct characteristics, our classification scheme facilitates the design and manufacture of prototypes.

The classification scheme also enables decision-making regarding the morphology inherent to each classification and the location of the necessary components for actuation. For instance, robots with endoskeletons (End) benefit from the distribution of actuation systems due to the presence of internal rigid components. In contrast, hydrostatic skeleton robots (HyS) lack rigid structures where components can be distributed and protected, necessitating the search for alternative methods compatible with the hydrostatic structure.

Furthermore, the classification scheme is helpful in defining the materials and procedures required for manufacturing soft robots. For example, a robot consisting of a muscular hydrostat and cable actuation (MuHMu) would require channels within a soft material for the passage of actuator cables, whereas a structure with an exoskeleton (Exo) can be correlated with additive manufacturing processes or even origami.

Figure 16 reveals that the majority of the 135 reviewed articles pertain to hydrostatic skeletons with inflatable or muscular actuation. This observation is not surprising given that hydrostatic skeletons, particularly those in the octopus, serve as a strong source of bioinspiration in soft robotics. This analysis further identifies areas of opportunity for soft robotics research. Specifically, the less common classifications include endoskeletons with muscular actuation, exoskeletons with inflatable actuation, and muscular hydrostats with inflatable actuation. By recognizing these less explored areas, researchers can concentrate on advancing soft robotics in novel directions.

## 5. Discussion and Conclusions

The advantage of this classification is discussed in Section 1 and Section 4. The proposed classification allows for an analytical analogy with other types of robots that are widely studied. For example, while a robot arm can be classified as *RRR* due to its three rotational joints, a soft muscle origami robot will be classified as ExoMu. In the same way that the kinematic model of a rotational joint in a robot arm is modeled using angular quantities, the rotational joints of an origami robot could also be modeled using angles. Similarly, the classification of HySIn soft robots, which are hydrostatic skeletons with inflatable actuation, could have kinematic and dynamic models based on fluid pressure. Dynamic models for MuHIn robots, which are muscular hydrostats with pneumatic actuators, could be based on fluid pressure and curve bending profiles, and so forth with all other classes.

The vast majority of animals on the planet use a support structure with adequate characteristics to live on land, sea, or air. Few organisms, such as octopuses and squids, have this apparent lack of support structure. As shown in Section 3, these invertebrates have a skeletal system that directly depends on the aqueous environment where they live. If these animals are taken to a different environment, they cannot move in the same way as in water. Therefore, soft robots that are designed to work in environments similar to those of humans, should probably have a morphology similar to that of other animals on the surface.

Ideally, all components of a soft robot should be soft, including power sources, sensors, and electronics. However, as of now, this is not yet possible. It is essential to note that the proposed classification scheme does not require a robot to be completely soft. The classification allows for the presence of rigid internal or external parts as long as the structural and actuation elements are soft. Robots that are rigid and covered with soft materials are excluded.

The muscular and inflatable actuator classes are designed to be very general. However, other types of muscles exist, as discussed in Section 3.2. Hence, the term “muscle” provides room for expansion and allows for more dimensions to be added to the classification. The same applies to inflatable actuation, which can be subdivided into different types of movements caused by the fluid, depending on cavity geometry, characteristics, or the work fluid. We believe that the flexibility of the classification will be advantageous for future expansions and subdivisions.

Using this bio-inspired classification, many hybrid options for classification are possible. For example, a hypothetical tortoise robot could be considered a soft robot type EndExoMu, which refers to a soft robot with an endoskeleton and exoskeleton that uses muscles for actuation. Another example is Disney’s Baymax from the movie Big Hero 6 [168], which is a soft robot-type EndIn until Hiro Hamada adds an exoskeleton, making it an EndExoIn soft robot. Jamming robots are another example of hybrid robots. They have soft structures until a vacuum is applied, turning their structure into a hydrostatic skeleton. Then, when actuated, they transform into an endoskeleton, as seen in [41] or the soft robot platform in [169] that uses jamming and rotational actuators. Some robots have two types of actuators that allow them to perform combined movements and actions, such as [170]. The use of different types of actuators and structures in soft robots creates interesting skills that increase capabilities and provide versatility in their actuation and movement.

This article can begin a discussion about the possible features of a classification method. Our hope is that the scientific community takes this proposal and improves it, remakes it, or discards it to make way for a better version that can guide the field of soft robotics toward the bright future of human–machine interaction.

## Figures and Tables

**Figure 1 biomimetics-08-00192-f001:**
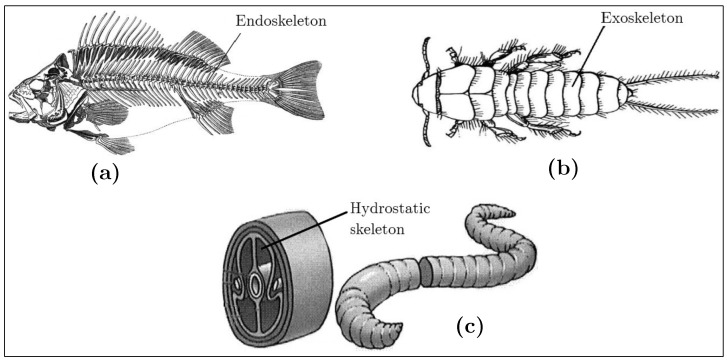
Skeletal animal structures. Vertebrate: (**a**) fish; invertebrates: (**b**) insects and (**c**) earthworms.

**Figure 2 biomimetics-08-00192-f002:**
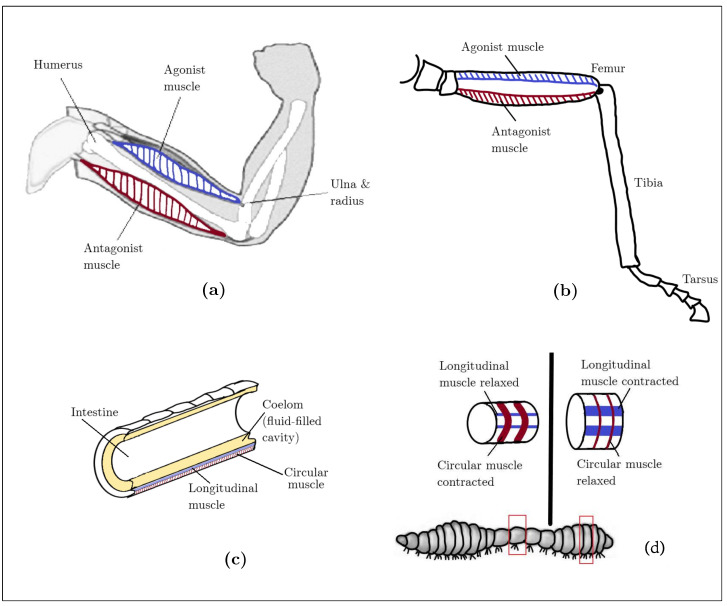
Types of animal actuation. (**a**) Agonist/antagonist muscle actuation with endoskeleton; (**b**) agonist/antagonist muscle actuation with exoskeleton; (**c**) hydrostatic skeleton actuation; (**d**) earthworm radial and longitudinal muscle actuation.

**Figure 3 biomimetics-08-00192-f003:**
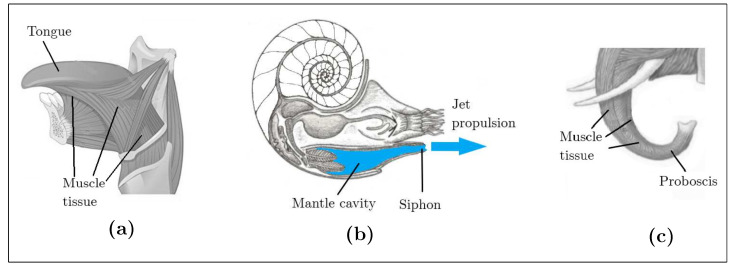
Animal support structures with actuation. Muscular hydrostat: (**a**) human tongue and (**b**) elephant trunk; animal inflation: (**c**) nautilus jet propulsion.

**Figure 4 biomimetics-08-00192-f004:**
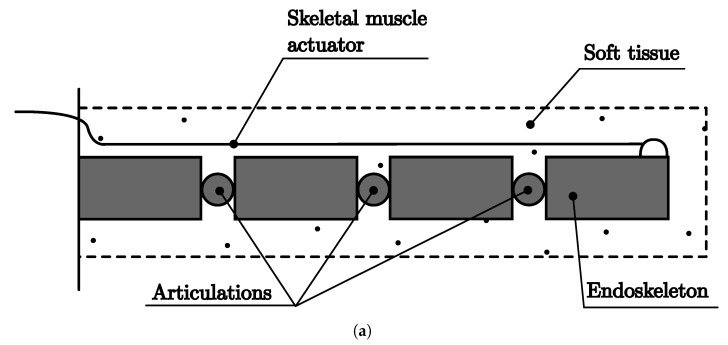
Soft robot EndMu Class. (**a**) Soft robot schematic diagram EndMu. (**b**) Soft robot prototype EndMu.

**Figure 5 biomimetics-08-00192-f005:**
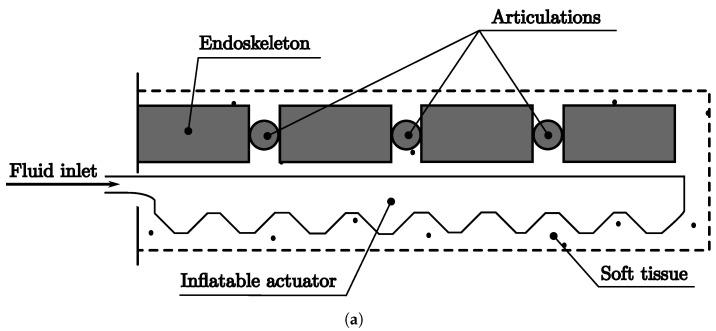
Soft robot EndIn Class. (**a**) Soft robot schematic diagram EndIn. (**b**) Soft robot prototype EndIn.

**Figure 6 biomimetics-08-00192-f006:**
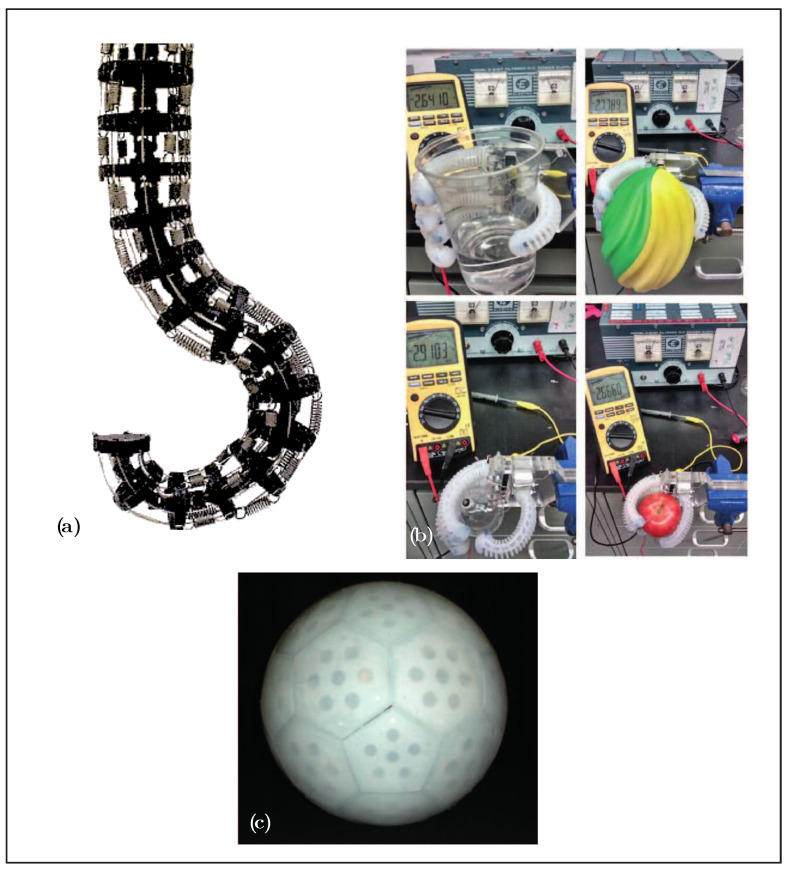
Examples of endoskeletons: (**a**) elephant’s trunk manipulator [31]; (**b**) soft humanoid robotic hand [32]; (**c**) spherical rolling robots [33], adapted with permission.

**Figure 7 biomimetics-08-00192-f007:**
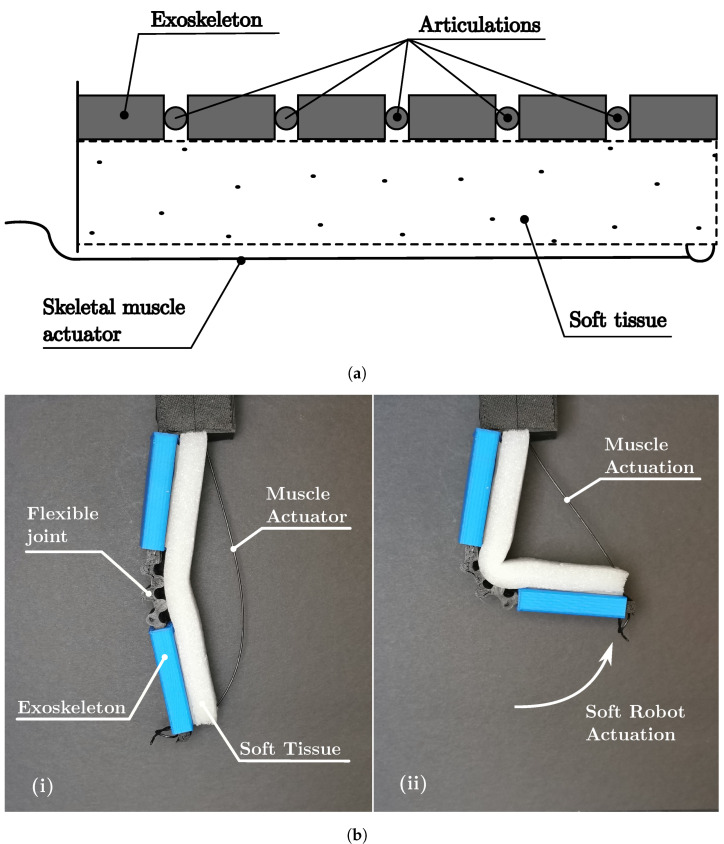
Soft robot ExoMu class. (**a**) Soft robot schematic diagram ExoMu. (**b**) Soft robot prototype ExoMu.

**Figure 8 biomimetics-08-00192-f008:**
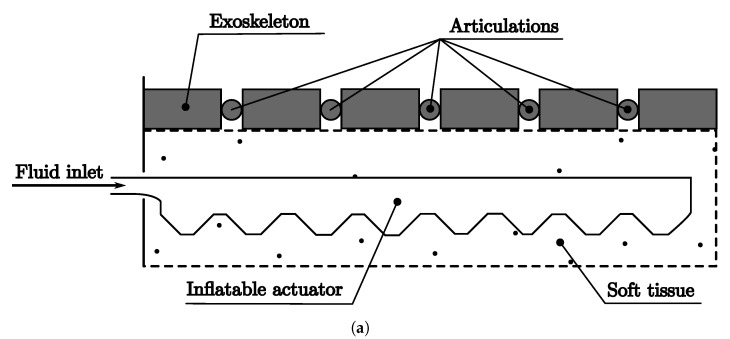
Soft robot ExoIn Class. (**a**) Soft robot schematic diagram ExoIn. (**b**) Soft robot prototype ExoIn.

**Figure 9 biomimetics-08-00192-f009:**
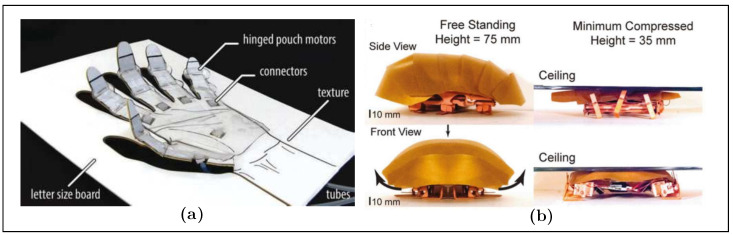
Examples of exoskeleton configurations: (**a**) origami robot with soft actuator [48]; (**b**) cockroach-inspired robot in a reduced space [49], adapted with permission.

**Figure 10 biomimetics-08-00192-f010:**
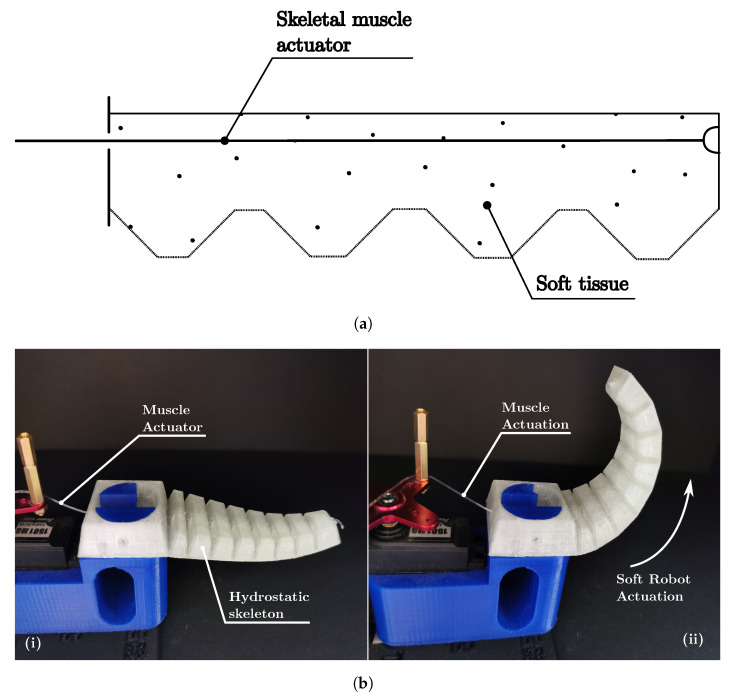
Soft robot, HysMu class. (**a**) Soft robot schematic diagram, HySMu. (**b**) Soft robot prototype, HySMu.

**Figure 11 biomimetics-08-00192-f011:**
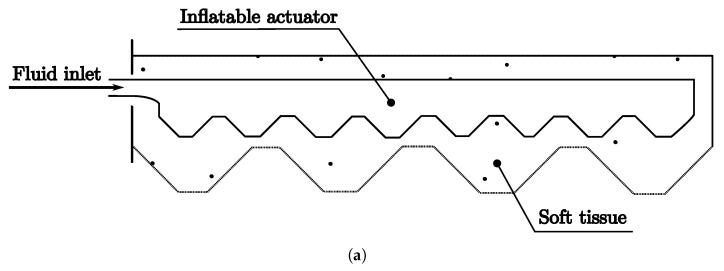
Soft robot, HySIn Class. (**a**) Soft robot schematic diagram, HySIn. (**b**) Soft robot prototype, HySIn.

**Figure 12 biomimetics-08-00192-f012:**
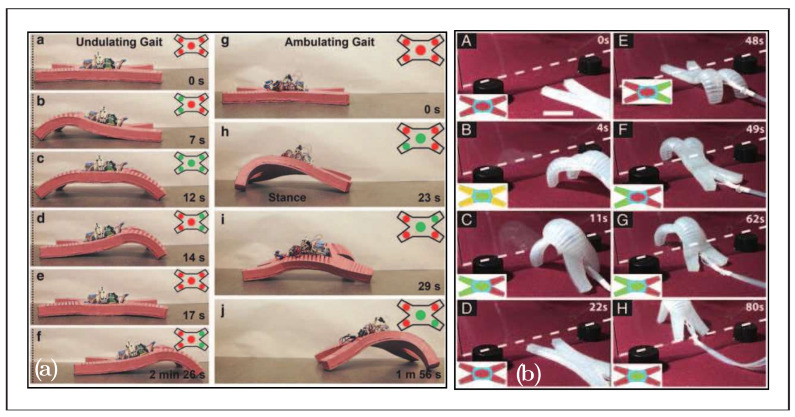
Examples of hydrostatic skeletons: (**a**) quadruped soft robot [76]; (**b**) quadruped soft robot [77], adapted with permission.

**Figure 13 biomimetics-08-00192-f013:**
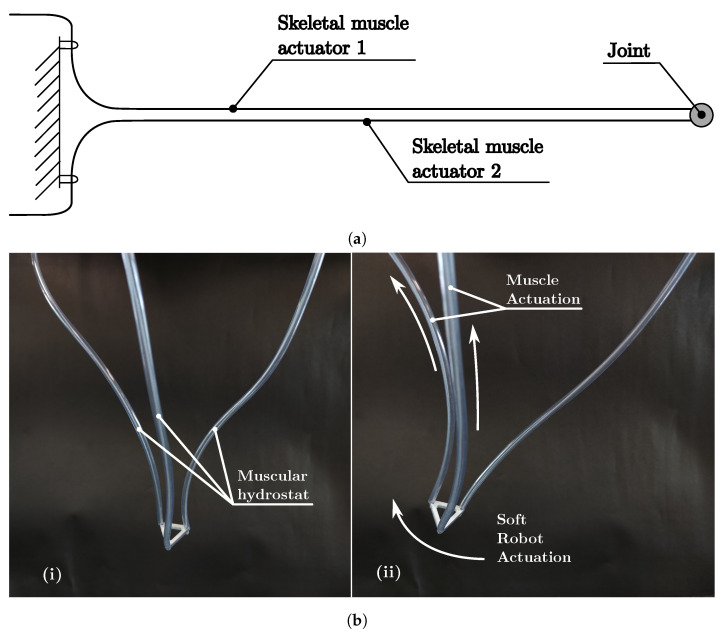
Soft robot MuHMu class. (**a**) Soft robot schematic diagram MuHMu. (**b**) Soft robot prototype MuHMu.

**Figure 14 biomimetics-08-00192-f014:**
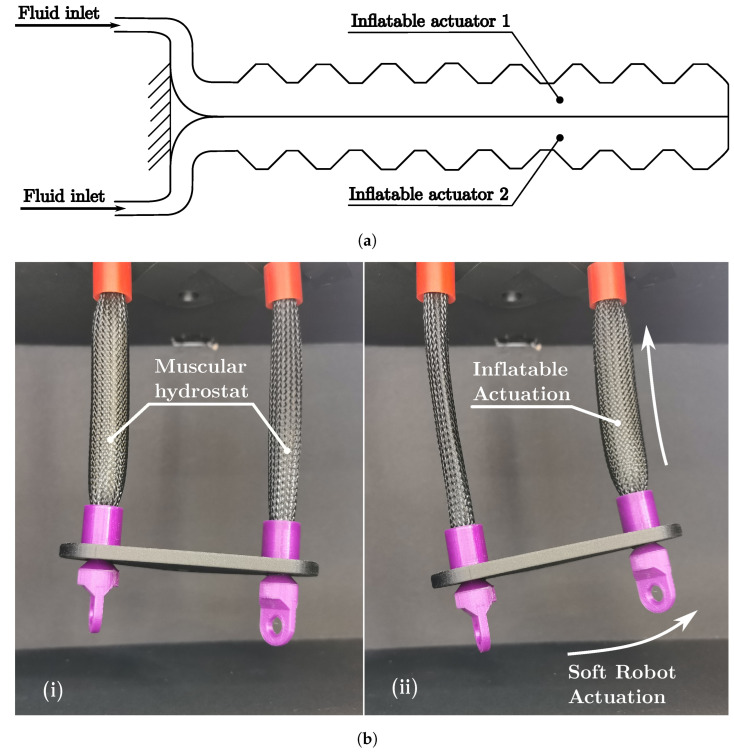
Soft robot MuHIn Class. (**a**) Soft robot schematic diagram MuHIn. (**b**) Soft robot prototype MuHIn.

**Figure 15 biomimetics-08-00192-f015:**
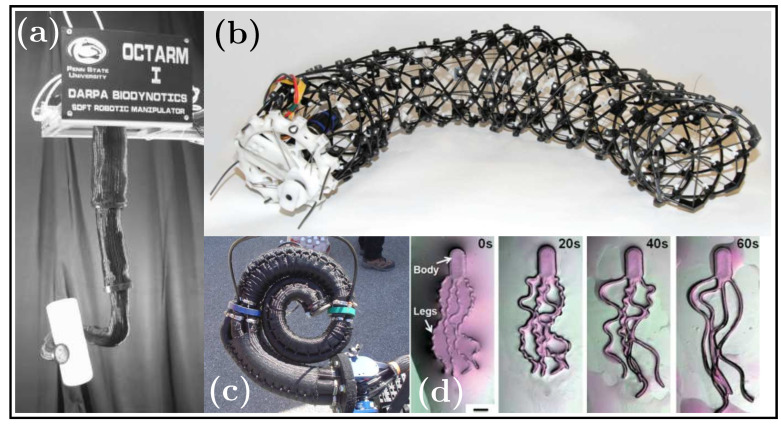
Examples of muscular hydrostats: (**a**) OctArm I [150]; (**b**) robot with peristalsis [152]; (**c**) OctArm V [151]; (**d**) Mini soft aquabots [153], adapted with permission.

**Figure 16 biomimetics-08-00192-f016:**
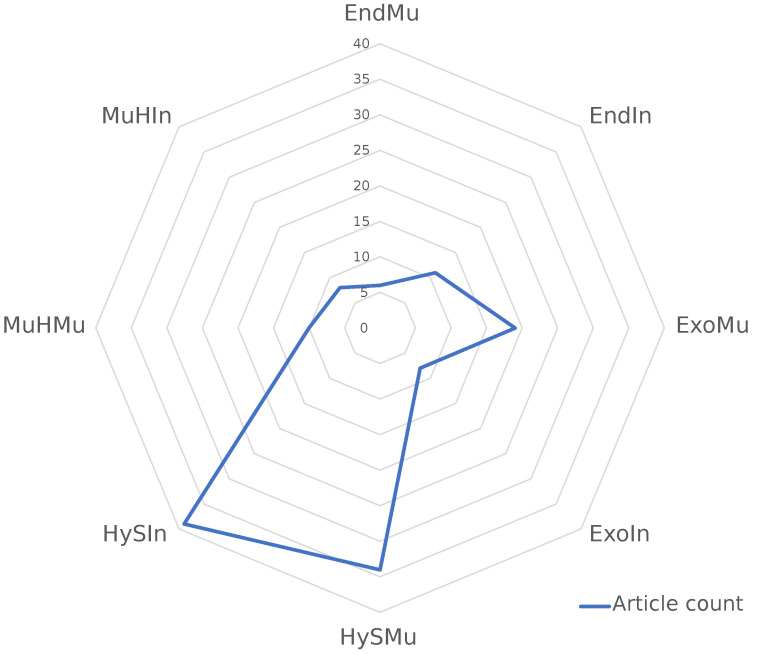
Classification graph of the 135 reviewed articles.

**Table 1 biomimetics-08-00192-t001:** Bio-inspired classification.

Support Structure	Class	Actuator Type	Class
Endoskeleton	End	Skeletal muscle	Mu
Exoskeleton	Exo	Inflatable	In
Hydrostatic skeleton	HyS		
Muscular hydrostat	MuH		

**Table 2 biomimetics-08-00192-t002:** Soft robot prototypes based on the proposed classification.

Soft Robot Type	Support Structure	Actuator Type	Figure	Table
EndMu	Endoskeleton	Skeletal muscle	Figure 4	Table 3
EndIn	Endoskeleton	Inflatable	Figure 5	Table 3
ExoMu	Exoskeleton	Skeletal muscle	Figure 7	Table 4
ExoIn	Exoskeleton	Inflatable	Figure 8	Table 4
HySMu	Hydrostatic skeleton	Skeletal muscle	Figure 10	Table 5
HySIn	Hydrostatic skeleton	Inflatable	Figure 11	Table 5
MuHMu	Muscular hydrostat	Skeletal muscle	Figure 13	Table 6
MuHIn	Muscular hydrostat	Inflatable	Figure 14	Table 6

**Table 3 biomimetics-08-00192-t003:** Classified soft robots with denomination endoskeleton.

Robot	Authors	Class.	Characteristics
iSprawl	Kim et al. [35]	EndMu	Cable-driven
Robot with amoeboid movement	Kaufhold et al. [36]	EndMu	Magnetic based actuators
Elephant’s trunk manipulator	Hannan et al. [31]	EndMu	Cable Driven, extension springs actuators
Soft humanoid robotic hand	She et al. [32]	EndMu	SMA actuator and PZT flexure sensor
Soft lattice modules	Zhao et al. [37]	EndMu	Shape memory alloy spring actuators
Tensegrity manipulator with tetrahedral parallel modules	Ramadoss et al. [38]	EndMu	Cable driven
Bio-inspired manipulator for MSI	Ranzani et al. [39]	EndIn	Pneumatic actuator granular jamming
Simple passive universal gripper	Amend et al. [40]	EndIn	Pneumatic actuator granular jamming
JSEL	Steltz et al. [41]	EndIn	Jamming skin Pneumatic actuator
STIFF-FLOP Surgical manipulator	Cianchetti et al. [42]	EndIn	Pneumatic actuators
Peristaltic crawling robot	Nakamura et al. [34]	EndIn	Pneumatic actuator
Spherical rolling robots	Wait et al. [33]	EndIn	Pneumatic actuator
Fluid-driven origami-inspired artificial muscles	Li et al. [43]	EndIn	Vacuum actuator
Multi-material metacarpophalangeal joint	Gollob et al. [44]	EndIn	McKibben muscles
Soft pneumatic modules	Nilles et al. [45]	EndIn	Pneumatic actuator
Hybrid jamming SR Fingers	Yang et al. [46]	EndIn	Hybrid Jamming and pneumatic actuator
RoBoa	Maur et al. [47]	EndIn	Pneumatic actuator

**Table 4 biomimetics-08-00192-t004:** Classified soft robots with denomination 2, exoskeleton.

Robot	Authors	Class.	Characteristics
Self-folding origami robot	Kim et al. [50]	ExoMu	Torsion shape memory alloy wire
Sprawlita	Cham et al. [51]	ExoMu	SDM-mechanical actuator
Origami wheel transformer	Lee et al. [52]	ExoMu	Coil spring actuator
Insect model-based microrobot	Suzuki et al. [53]	ExoMu	Electrostatic actuator
PAC hinge	Ge et al. [54]	ExoMu	Thermomechanical actuator
Soft adaptive robotic fish	Liu et al. [55]	ExoMu	Cable-driven actuator
Crawling robot	Pagano et al. [56]	ExoMu	Motor-driven actuator
2 DOF hexapod	Faal et al. [57]	ExoMu	Motor-driven actuator
Self-folding robot	Felton et al. [58]	ExoMu	Motor-driven actuator
Robogami	Firouzeh et al. [59]	ExoMu	SMA torsional actuator
Origami-inspired worm robot	Onal et al. [60]	ExoMu	Nickel titanium coil actuators
Omega-shaped inchworm	Koh et al. [61]	ExoMu	SMA coil-spring actuator
Self-folding crane	Felton et al. [62]	ExoMu	Resistive circuits actuators
Origami water bomb-based	Bowen et al. [63]	ExoMu	MagnetoActive Elastomer Actuators
Helical Kirigami	Zhang et al. [64]	ExoMu	Linear SMA Actuators
OrigamiBot-II: Three-finger origami manipulator	Jeong et al. [65]	ExoMu	Servomotors actuator
Quad-Spatula gripper	Gafer et al. [66]	ExoMu	Cable driven actuators
Push puppet soft-rigid robot	Bern et al. [67]	ExoMu	Cable-driven actuators
Salamanderbot	Sun et al. [68]	ExoMu	Cable-driven actuators
Robot Jumper	Bartlett et al. [69]	ExoIn	Butane/oxygen combustion
Soft robot that can imitate an earthworm	Zhou et al. [70]	ExoIn	Pneumatic actuator
Soft biomimetic prosthetic hand	Fras et al. [71]	ExoIn	Pneumatic actuator
Planar-printable robotic hand	Niiyama et al. [48]	ExoIn	Pouch Motor Pneumatic Actuator
Soft robotic bladder array	Aston et al. [72]	ExoIn	Pneumatic actuators
Otariidae-inspired SR	Liu et al. [73]	ExoIn	Pneumatic actuators
VPAM	Zhang et al. [74]	ExoIn	Pneumatic actuators
Hybrid soft robot	Archchige et al. [75]	ExoIn	Pneumatic actuators

**Table 5 biomimetics-08-00192-t005:** Classified soft robots with denomination 3, hydrostatic skeleton.

Robot	Authors	Class.	Characteristics
Meshworm	Seok et al. [79]	HySMu	Nickel titanium (NiTi) coil actuators
Octopus Robots	Cianchetti et al. [80]	HySMu	Cable driven actuators and SMA
Robot arm	Cheng et al. [81]	HySMu	Cable driven actuators
Robot worm	Lin et al. [82]	HySMu	SMA coil
Robot arm	Laschi et al. [83]	HySMu	SMA coil
Stickybot	Kim et al. [84]	HySMu	Push–pull cable actuator
Artificial octopus muscle	Follador et al. [85]	HySMu	SMA springs
PoseiDRONE	Arienti et al. [86]	HySMu	Cable-driven actuator
Softworms	Umedachi et al. [87]	HySMu	SMA coils and motor tendons
Undulating body by SMA	Low et al. [88]	HySMu	SMA actuator
A 3D-printed soft robot	Umedachi et al. [89]	HySMu	3-D Printed, SMA actuators
Spherical deformable robot	Sugiyama et al. [90]	HySMu	SMA coils and polymer gel actuators
Soft robot for cardiac ablation	Deng et al. [91]	HySMu	Cable driven actuator
Fish-like underwater microrobot	Guo et al. [92]	HySMu	ICPF actuator
Robotic cownose ray microrobot	Chen et al. [93]	HySMu	IPMC actuator
Soft robot mimics caterpillar locomotion	Rogóż et al. [94]	HySMu	LCE film with patterned molecular orientation, light-driven
SDM hand	Dollar et al. [95]	HySMu	Cable driven actuator
Jellyfish 2D muscle architecture	Nawroth et al. [78]	HySMu	Bio-hybrid actuators
Plastic frame shell dielectric elastomer actuator	Kofod et al. [96]	HySMu	Dielectric elastomer actuator.
Soft frog robot	Su et al. [97]	HySMu	Multilayer composite of SMP and polyurethane, SMA actuator
Loco-sheet	Chang et al. [98]	HySMu	Cable-driven actuator
Quadrupedal starfish soft robot	Munadi et al. [99]	HySMu	String-driven actuator
Soft finger with a pneumatic sensor	Tawk et al. [100]	HySMu	Cable-driven actuator, pneumatic sensor
Soft robotic fingers	Teeple et al. [101]	HySIn	Pneumatic actuator
Cable-driven soft gripper	Honji et al. [102]	HySMu	Cable-driven actuator
Robotic jellyfish	Gatto et al. [103]	HySMu	Cable-driven actuator
Soft SMA-powered limb	Patterson et al. [104]	HySMu	SMA actuators
Planar soft robot	Zheng et al. [105]	HySMu	Piezoelectric actuator
Untethered soft millirobot	Bhattacharjee et al. [106]	HySMu	Magnetic actuator
Plant tendril-like soft robot	Meder et al. [107]	HySMu	Heating element actuator
Climbing soft robot	Sakuhara et al. [108]	HySMu	Cable driven actuator
Electrostatic/gecko-inspired SR	Alizadehyazdi et al. [109]	HySMu	Cable-driven actuator
Untethered soft robot	Oh et al. [110]	HySMu	Heating element actuator
Inchworm–earthworm-like Soft Robots	Karipoth et al. [111]	HySMu	Magnetic actuator
TENG-Bot	Sun et al. [112]	HySMu	Dielectric actuator
Quadruped soft robot	Tolley et al. [76]	HySIn	Pneumatic actuator
Quadruped soft robot	Shepherd et al. [77] Morin et al. [113]	HySIn	Pneumatic actuators
Robotic fish	Marchese et al. [114]	HySIn	Pneumatic actuators
Tripedal soft robot	Shepherd et al. [115]	HySIn	Methane/oxygen combustion
Actuator that actuates rapidly	Mosadegh et al. [116]	HySIn	Pneumatic actuator
Bio-inspired soft robotic snake	Onal et al. [117]	HySIn	Pneumatic actuator
Soft mobile-rolling robot	Onal et al. [118]	HySIn	Catalyzed decomposition of hydrogen peroxide
Untethered jumping soft robot	Tolley et al. [119]	HySIn	Pneumatic and chemical (Butane combustion) actuator
Four-legged quadruped	Stokes et al. [120]	HySIn	Pneumatic actuators
Manta swimming robot	Suzumori et al. [121]	HySIn	Pneumatic actuators
Soft robot for thumb rehabilitation	Maeder-York et al. [122]	HySIn	Pneumatic actuator
Octopus-inspired suction cups	Follador et al. [123]	HySIn	Dielectric elastomer actuator
The second skin: soft robot assistive device	Goldfield et al. [124]	HySIn	Soft pneumatic synthetic muscles and strain sensors
RBO hand 2	Deimel et al. [125]	HySIn	Pneumatic actuators
Multi-fingered robot arm	Suzumori et al. [126]	HySIn	Flexible microactuators driven with pneumatics
Deformable 2D robotic manipulation system	Marchese et al. [127]	HySIn	Pneumatic Actuator
Soft robotic glove	Polygerinos et al. [128]	HySIn	Pneumatic actuators
Robotic tentacles	Martinez et al. [129]	HySIn	Pneumatic actuators
Soft wearable robotic device for ankle-–foot rehabilitation	Park et al. [130]	HySIn	Pneumatic actuators
PneuArm	Sanan et al. [131]	HySIn	Pneumatic torsional actuators
Soft left ventricle	Roche et al. [132]	HySIn	McKibben actuators
Multi-fingered soft robotic hand	Devi et al. [133]	HySIn	Pneumatic actuator
Soft robotic surface	Chen et al. [134]	HySIn	Pneumatic actuators
Soft robot with crawling locomotion	Qi et al. [135]	HySIn	Pneumatic actuators
Soft Modular Robotic Cubes	Vergara et al. [136]	HySIn	Pneumatic actuator, magnetic modular cubes
Soft robot for granular media	Ortiz et al. [137]	HySIn	Pneumatic actuator
Soft robotic fingers	Truby et al. [138]	HySIn	Pneumatic actuator
Inchworm crawling	Gamus et al. [139]	HySIn	Pneumatic actuator
Soft robot with peristaltic movement	Das et al. [140]	HySIn	Pneumatic actuator
EELWORM	Milana et al. [141]	HySIn	Pneumatic actuator
Flexible connector for soft modular robots	Tse et al. [142]	HySIn	Pneumatic actuator
PRR	Partridge et al. [143]	HySIn	Pneumatic actuator
Soft actuator	Yao et al. [144]	HySIn	Pneumatic actuator
Hip abduction actuator	Yang et al. [145]	HySIn	Pneumatic actuator
Enveloping soft gripper	Hao et al. [146]	HySIn	Pneumatic actuator
Soft wearable exoskeleton	Ma et al. [147]	HySIn	Pneumatic actuator
Starfish-like soft robot	Zou et al. [148]	HySIn	Pneumatic actuator
Dexterous soft robotic hand	Abondance et al. [149]	HySIn	Pneumatic actuators

**Table 6 biomimetics-08-00192-t006:** Classified soft robots with denomination 4, muscular hydrostat.

Robot	Authors	Class.	Characteristics
X-RHex	Galloway et al. [154]	MuHMu	DC motor to drive the slider
Robot with peristalsis for locomotion	Boxerbaum et al. [152]	MuHMu	Bouden cable
IPMC-patterned actuator	Nakabo et al. [155]	MuHMu	IPMC Actuator
Mini soft aquabots	Kwon et. al. [153]	MuHMu	Electroactive hydro-gel-based actuators.
Starfish gel robot	Otake et al. [156]	MuHMu	Hydrogel electro-actuator
Gel walker	Morales et al. [157]	MuHMu	Hydrogel electro-actuator
Soft Auxiliary Arm	Yu et al. [158]	MuHMu	Cable-driven actuator
Tendon-driven continuum mechanism	Deutschmann et al. [159]	MuHMu	Tendon driven
Triboelectric soft robot	Liu et al. [160]	MuHMu	Triboelectric actuators
Piecewise controllable soft robots	Li et al. [161]	MuHMu	Electrothermal actuator
Robot jumper	Walker et al. [150]	MuHIn	Pneumatic (McKibben)
OctArm V-continuum manipulator	McMahan et al. [151]	MuHIn	Pneumatic Actuator
Soft fluidic elastomer manipulator	Marchese et al. [162]	MuHIn	Pneumatic actuator
Soft Arm	Rafter et al. [163]	MuHIn	Pneumatic actuator
Soft robot Kaa	Bodily et al. [164]	MuHIn	Pneumatic actuator
Peristaltic continuous mixing conveyor	Wakamatsu et al. [165]	MuHIn	Pneumatic actuator
Untethered knit fabric soft robot	Nguyen et al. [166]	MuHIn	Pneumatic actuator
Pneumatic bending actuators	Lamping et al. [167]	MuHIn	Pneumatic actuator

## Data Availability

The data that support the findings of this study are available from the corresponding author, A. López-Gonzalez (alexandro.lopez@ibero.mx), upon reasonable request.

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
