# Peer review of "Review and Proposal for a Classification System of Soft Robots Inspired by Animal Morphology"

_biomimetics, 2023, doi:10.3390/biomimetics8020192_

Round 1

Reviewer 1 Report

The authors present an excellent and timely review article where they propose a classification strategy for bioinspired soft robotics. The authors rationalize that while the field of soft robotics has undergone a lot of progress over the past decade, different categories of bioinspired soft robots remain disorganized. As a result, the field is in dire need of a formal classification.

The paper is well written overall, but will probably require some professional English language editing to make it more readable. Apart from this, I have only a few comments, which could further improve the manuscript:

1.     Each figure that is not original and adapted from another source, should indicate a reference to the source in the caption with something to the tune of “Reproduced/Adapted with permission”.

2.     The captions of the figures are currently very minimal and lacking in context. More descriptive captions will be helpful to the reader and complement the text.

3.     I commend the authors on creating their own soft robot prototypes for this manuscript – even though it is a review. If they could provide engineering schematics showing the internal structure of the fabricated soft robots, that would be beneficial.

4.     The authors do an excellent job in classifying all published soft robot work according to their proposed strategy. What would make this classification more easily digestible at first glance without going through the long tables is if the authors can provide some form of meta-analysis of the published papers – graphical representation number of papers in each class, characteristic, most common class/characteristic, etc.

Author Response

Juan C. Tejada

Faculty of Engineering

Computational Intelligence and Automation Group (GIICA)

EIA University

Km2+200 ms. Variante Aeropuerto José María Córdova

Envigado, Antioquia

Email: juan.tejada@eia.edu.co

March 30, 2023

Re: Submission of revised version of manuscript

Authors response to Reviewers Comments

 Title: “Review and proposal of a Soft Robots classification inspired on animal morphology”

We would like to thank the reviewer for the comments and suggestions to improve the paper. The changes made in the manuscript were highlighted to ease the revision of the paper.

Comment 1: Each figure that is not original and adapted from another source, should indicate a reference to the source in the caption with something to the tune of “Reproduced/Adapted with permission”.

 Answer 1:

The text has been added.

Comment 2: The captions of the figures are currently very minimal and lacking in context. More descriptive captions will be helpful to the reader and complement the text.

Answer 2: The captions of the figures have been improved.

Comment 3: I commend the authors on creating their own soft robot prototypes for this manuscript – even though it is a review. If they could provide engineering schematics showing the internal structure of the fabricated soft robots, that would be beneficial.

Answer 3: The schematic diagram has been added to each soft robot class.

Comment 4: The authors do an excellent job in classifying all published soft robot work according to their proposed strategy. What would make this classification more easily digestible at first glance without going through the long tables is if the authors can provide some form of meta-analysis of the published papers – graphical representation number of papers in each class, characteristic, most common class/characteristic, etc.

Answer 4: The section “4.3 Classification analysis” has been added to the paper to support the proposal classification. In this section the benefits and characteristics about the classification are discussed. A classification graph of the reviewed articles has been added to improve and to easily the discussion.

The reviewer is again thanked for the comments, and any questions or concerns that may arise regarding the review, the authors will be quick to respond. We hope that our revised paper meets with your satisfaction.

Sincerely yours,

Alexandro López-González, Juan C. Tejada and Janet López-Romero

Reviewer 2 Report

The work by López-González et al selected a nice perspective to review soft robots. it should be well matched to the readers interest.  However, the analysis is a bit superficial. More discussions and outlook should be given.

major comments:

1,  the reasons why the authors cauterized the research should be well addressed. what kind of new perspective will be given to readers, how will the readers will benefit from you new calcifications

2,  what are the main features for your sub-categorization? advantages, disadvantages for robot design, fabrication, or control; or new applications? better to give a clear comparison of them

3, any new opportunities for the researchers if following your opinions?

4, the comments features of bio-inspired designs and corresponding creatures'  should be well defined

minors

1, Figure 1, Does earthworm belong to vertebrate?

2, Fig 2, no general caption. does it mean the subfigures are not relevant, and should be put on by on instead of in one figure?

Author Response

Juan C. Tejada

Faculty of Engineering

Computational Intelligence and Automation Group (GIICA)

EIA University

Km2+200 ms. Variante Aeropuerto José María Córdova

Envigado, Antioquia

Email: juan.tejada@eia.edu.co

March 30, 2023

Re: Submission of revised version of manuscript

Authors response to Reviewers Comments

Title: “Review and proposal of a Soft Robots classification inspired on animal morphology”

We would like to thank the reviewer for the comments and suggestions to improve the paper. The changes made in the manuscript were highlighted to ease the revision of the paper.

Comment 1: The reasons why the authors cauterized the research should be well addressed. what kind of new perspective will be given to readers, how will the readers will benefit from you new calcifications

Answer 1: To clarify this comment the next paragraph have been added to the text. In the section “1. Introduction”:

 Soft robotics is an emerging field with a wide range of materials, platforms, actuators, sensors, and control strategies. As such, it can be challenging to navigate and organize the diverse advancements made in the field, resulting in a fragmented and disorganized landscape. However, the authors of this work propose a solution to this problem in section 4. Is suggested that a bio-inspired classification could be used to structure the study area of soft robotics, which would help to systematize the various advances in the field and generate a better understanding of the science behind the technology.

The proposed bio-inspired classification scheme has the potential to bring order to the field of soft robotics and facilitate the development of new and innovative soft robotic technologies. By grouping advances into structure and actuation categories based on their morphology, researchers can more easily identify areas of opportunity for further exploration and development. This classification scheme may also allow for a more straightforward comparison of different soft robotic technologies, enabling researchers to assess their relative strengths and weaknesses in a more meaningful way.

Additionally, the section “4.3 Classification analysis” has been added to the paper to support the proposal classification. In this section the benefits and characteristics about the classification are discussed. A classification graph of the reviewed articles has been added to improve and to easily the discussion:

The proposed classification scheme offers an overview of the development of soft robots, providing a clear and concise way to define the requirements and characteristics of different types of soft robots. In engineering design processes, properly defining the requirements and characteristics is essential, as without it, the range of possibilities can become overwhelming. By establishing four types of soft robots with distinct characteristics, our classification scheme facilitates the design and manufacture of prototypes.

The classification scheme also enables decision-making regarding the morphology inherent to each classification and the location of the necessary components for actuation. For instance, robots with endoskeletons (End) benefit from the distribution of actuation systems due to the presence of internal rigid components. In contrast, hydrostatic skeleton robots (HyS) lack rigid structures where components can be distributed and protected, necessitating the search for alternative methods compatible with the hydrostatic structure.

Furthermore, the classification scheme is helpful in defining the materials and procedures required for manufacturing soft robots. For example, a robot consisting of a muscular hydrostat and cable actuation (MuHMu) would require channels within a soft material for the passage of actuator cables, whereas a structure with an exoskeleton (Exo) can be correlated with additive manufacturing processes or even origami.

 Figure 16 reveals that the majority of the 135 reviewed articles pertain to hydrostatic skeletons with inflatable or muscular actuation. This observation is not surprising given that hydrostatic skeletons, particularly those in the octopus, serve as a strong source of bioinspiration in soft robotics. This analysis further identifies areas of opportunity for soft robotics research. Specifically, the less common classifications include endoskeletons with muscular actuation, exoskeletons with inflatable actuation, and muscular hydrostats with inflatable actuation. By recognizing these less explored areas, researchers can concentrate on advancing soft robotics in novel directions.

Comment 2: what are the main features for your sub-categorization? advantages, disadvantages for robot design, fabrication, or control; or new applications? better to give a clear comparison of them.

Answer 2: This comment has been answered in the new section “4.3 Classification analysis”.

Comment 3: any new opportunities for the researchers if following your opinions?

Answer 3: This comment has been answered in the new paragraph in the Introduction and the new section “4.3 Classification analysis”.

Comment 4: the comments features of bio-inspired designs and corresponding creatures'  should be well defined

Answer 4: To improve this situation, has been added schematics diagrams to each soft robot class. This makes this classification more easily digestible. Additionally, figures captions have been improved.

Minor comments:  1, Figure 1, Does earthworm belong to vertebrate?

2, Fig 2, no general caption. does it mean the subfigures are not relevant, and should be put on by on instead of in one figure?

 Answer minor comments: Both comments have been attended.

The reviewer is again thanked for the comments, and any questions or concerns that may arise regarding the review, the authors will be quick to respond. We hope that our revised paper meets with your satisfaction.

Sincerely yours,

Alexandro López-González, Juan C. Tejada and Janet López-Romero